# Vaccine Research Trends in Africa from 2016 to Mid-2024: A Bibliometric Analysis

**DOI:** 10.3390/vaccines13050509

**Published:** 2025-05-12

**Authors:** Chinwe Iwu-Jaja, Duduzile Ndwandwe, Thobile Malinga, Lindi Mathebula, Akhona Mazingisa, Charles Shey Wiysonge

**Affiliations:** 1Vaccine Preventable Diseases Programme, World Health Organization Regional Office for Africa, Brazzaville P.O. Box 06, Congo; sheyc@who.int; 2Cochrane South Africa, South African Medical Research Council, Cape Town 7505, South Africa; duduzile.ndwandwe@mrc.ac.za (D.N.); thobile.malinga@mrc.ac.za (T.M.); lindi.mathebula@mrc.ac.za (L.M.); 3Department of Community Health Studies, Faculty of Health Sciences, Durban University of Technology, Durban 4001, South Africa; akhonamazingisa04@gmail.com

**Keywords:** vaccine, immunization, vaccine trials, vaccine reviews, research, innovation, research productivity, Africa

## Abstract

Background: Vaccine research publications play a crucial role in the scientific process by strategically linking the generation of knowledge with its translation into vaccine policy and practice. This study was designed to understand vaccine and immunization research publication trends in Africa to inform strategic directions for vaccine research and innovation efforts in the continent. Methods: We searched PubMed only for vaccine and immunization-related publications from Africa between 1 January 2016 and 8 August 2024. Metrics such as annual growth rates, geographical distribution, international collaboration, and trend topics were analyzed. We conducted separate analyses for general vaccine research, vaccine clinical trials, and vaccine evidence syntheses (systematic reviews and meta-analyses). Results: Vaccine research in Africa demonstrated an annual growth rate of 55.4% (based on the 10,000 records retrieved due to PubMed’s export limit), while vaccine trials saw a decline of 6.08% during the study period. The trend topics analysis across vaccine research, trials, and reviews showed that topics shifted from a focus on general vaccine development, immunization, and malaria pre-2020 to COVID-19-related topics in 2020, with post-2020 research returning to traditional topics like immunization schedules, vaccine safety, and pediatric and maternal vaccines. Additionally, the COVID-19 pandemic had a profound impact on vaccine research, leading to a surge in publications for vaccine research, trials, and reviews. About 65.8% of vaccine research featured international co-authorship. Vaccine trials had a higher rate of international co-authorship at 79.8%. Conclusion: While vaccine research in general in Africa has increased, vaccine trials do not match this increase. The number of clinical trials remained relatively stagnant, reflecting ongoing challenges in the vaccine research ecosystem, particularly in building and sustaining clinical trial capacity across the region. In addition, disparities in research productivity exist between countries. Research prioritization, strategic collaborations, capacity building for research, and improved research infrastructure require critical consideration.

## 1. Introduction

Vaccines have consistently shown over several decades to be highly effective in the prevention and control of many infectious diseases, therefore supporting global health security. Despite this, the full potential of vaccines remains unrealized, particularly in Africa, where there is a disproportionately high burden of vaccine-preventable diseases and millions of children still miss out on vaccines yearly [1].

In recent years, increased attention has been directed toward developing new vaccines to combat various diseases. There has been significant innovation in vaccine development; new vaccines have been developed to combat diseases such as malaria, dengue, Ebola virus disease, and COVID-19. Promising vaccines against respiratory syncytial virus, tuberculosis, and various influenza virus strains are in the pipeline [1], marking a shift toward addressing Africa’s specific health needs [1]. African researchers and institutions are increasingly contributing to these innovations, driven by an understanding of the continent’s unique public health challenges. However, gaps remain, particularly in research infrastructure, funding, and, arguably, research prioritization.

The Immunization Agenda 2030 (IA2030) has set an ambitious goal of ensuring no one is left behind from receiving life-saving vaccines. Within the IA2030, research and innovation are identified as strategic priorities, as they potentially offer solutions to emerging and re-emerging disease threats, enhance preparedness and response strategies, improve access to vaccines, identify factors affecting immunization coverage, and provide interventions addressing inequities [1]. Additionally, conversations are ongoing about the need to enhance the research ecosystem in Africa as research is a driving force in advancing health care. African researchers are well informed about local contexts in Africa, making it essential for them to lead vaccine and immunization research efforts to address issues specific to Africa.

Bibliometric analysis, which uses publications as a proxy for research productivity, provides a valuable approach to assessing research trends, collaborations, and thematic focus areas [2,3]. A bibliometric analysis of vaccine research in Africa is crucial in understanding the current research landscape of Africa’s contributions to vaccine research worldwide, highlighting both successes and gaps. In addition, one global study focused solely on systematic reviews published between 2000 and 2017 [4], while another analyzed vaccine trials conducted in Africa from 2010 to 2024 [5]. While these studies provide valuable insights, to our knowledge, no bibliometric study has comprehensively analyzed general vaccine and immunization research, including trials and reviews, across Africa over the past decade. There is a need for a study that takes a broader view, as this is important for taking stock of where we are in the vaccine research landscape in the continent, assessing progress, and identifying any gaps.

We therefore conducted this bibliometric analysis of vaccine research conducted in Africa between 2016 and 2024, providing insights into key thematic areas and the level of collaboration between researchers locally and internationally, and highlighting the leading African countries in this field. Our study also provides critical data for evidence-informed planning, implementation, and monitoring and evaluation of immunization programs in the continent. As Africa aims to expand local vaccine production, this analysis will set the tone for addressing unfinished public health agendas, guiding research priorities, and supporting the capacity-building efforts necessary for a sustainable and equitable vaccine research ecosystem in the region.

## 2. Materials and Methods

In this study, a bibliometric analysis of vaccine related publications from all African countries between 2016 and 2024 was conducted. Our main source was the PubMed database, which was searched on 8 August 2024, to include publications from 1 January 2016 to 8 August 2024. These dates were chosen to correspond to the period of the transformation agenda of the WHO Regional Office for Africa (2015–2024), which aimed to enhance responsiveness and accountability in supporting Member States to improve health outcomes. This timeframe allowed for us to take stock of vaccine research trends across the region, highlighting areas of progress and identifying aspects where further support is needed within the framework of this agenda [6].

The PubMed database provides comprehensive coverage of the medical, biomedical, and health sciences literature. It indexes a wide range of document types, including original research articles, review articles, and clinical trials [7]. Additionally, it provides a functionality that allows for publications to be downloaded in a format that facilitates bibliometric analyses. We systematically searched the literature by combining key words and their synonyms using the appropriate Boolean operators, “OR” and “AND”. We used keywords such as vaccine, immunization, and Africa. To obtain country-specific publications, we listed all the countries in the African continent in our search strategy. This was helpful to ensure all countries were included and to precisely distinguish between countries with similar names (e.g., Congo versus Democratic Republic of Congo, Guinea versus Guinea-Bissau). Our literature search did not include any language restrictions.

We conducted searches at three levels. Firstly, we conducted a search to obtain publications on general vaccine and immunization research (which is referred to in this paper as vaccine research), comprising all types of research regardless of study type or design, including basic sciences, immunology, epidemiology, implementation research, biotechnology, randomized trials, literature reviews, etc. The second level was focused on vaccine and immunization related randomized trials, including phases I to IV clinical trials, referred to in this manuscript as vaccine trials. The third level consisted of systematic reviews and meta-analyses on vaccines and immunization conducted in Africa, which we refer to in this article as vaccine reviews. The detailed search strategies used for each of these searches are presented in Table A1, Table A2 and Table A3, respectively.

After conducting the search, the articles were retrieved, converted into a bibliographic data frame, and analyzed using the Bibliometrix R package [8]. As a first step, the publications were summarized into the number of articles retrieved, types (whether journals, books, etc.), article average age, and authors that published during the study period. Additionally, we determined the average number of co-authors per article and the percentage of all the retrieved articles that included international authors (that is, authors that are based outside of Africa). Annual growth rate (AGR) values were automatically computed using the Bibliometrix R package (R version 4.4.1). The AGR for general vaccine research was computed using the 10,000 records imported into Bibliometrix, owing to the PubMed export limit. However, AGRs for vaccine trials and reviews were derived from their full retrieved datasets, which were not constrained by this limitation.

Next, we determined the annual scientific productions, which is the number of articles published annually within the study duration. Then we compiled the list of the top prolific authors and institutions, top productive countries, the geographical distribution of countries’ productivity, and an analysis of trend topics [9]. Due to software constraints in standardizing country names, certain countries with similar or overlapping spellings (e.g., Congo and Democratic Republic of Congo; Guinea and Guinea-Bissau) were merged during analysis, resulting in combined article counts.

## 3. Results

### 3.1. Overview of the Bibliometric Analysis

A total of 17,169 articles were retrieved from the literature search on vaccine research. However, only the first 10,000 articles (starting from 1 January 2016) were included in the analysis for vaccine research due to PubMed’s retrieval limit. For vaccine trials and vaccine reviews, we retrieved and analyzed all publications, as the total count was less than 10,000 in each instance (Table 1).

Vaccine research showed a strong annual growth rate of 55.39%, while vaccine trials declined by 6.08%, and reviews grew by 20.98%. A total of 52,848 authors contributed to vaccine research, 5866 to trials, and 3814 to reviews, with high levels of international co-authorship, particularly for vaccine trials (79.84%) (Table 1).

### 3.2. Annual Research Trends

#### 3.2.1. Annual Scientific Production for Vaccine Research Landscape in Africa (2016–2024)

An upward trend in the number of publications for vaccine research in Africa was evident from our findings. Starting from 1000 publications in 2016, there was a consistent increase each year, with a more pronounced increase from 2020 onwards. The peak was reached in 2022 with 2432 publications, followed by a slight decline in 2023. Overall, there was nearly a two-fold increase in the number of publications between 2016 and 2024 (Figure 1).

#### 3.2.2. Annual Scientific Production for Vaccine Trials in Africa (2016–2024)

The graph in Figure 1 shows the yearly publications for vaccine trials in Africa from 2016 to 2024. A downward trend in number of publications for vaccine trials was observed from 2016 to 2018. Thereafter, there was an upward trend, reaching its peak in 2022, followed by a sharp decrease in 2023.

#### 3.2.3. Annual Scientific Production for Vaccine Reviews in Africa (2016–2024)

There was a steady increase in publications between 2016 and 2019, followed by a sharp rise to 2022, then a mild increase to peak in 2023 (Figure 1).

### 3.3. Most Prolific Institutions and Authors for Vaccine Research, Trials, and Reviews

#### 3.3.1. Most Prolific Institutions and Authors for Vaccine Research in Africa

Figure 2 shows the top 20 prolific institutions and authors, respectively.

The bar graph displays the top 20 institutions for vaccine research from 2016 to 2024, ordered by the number of publications. The African institution with the most publications was the University of Cape Town, followed by the University of the Witwatersrand and the University of KwaZulu Natal, all located in South Africa. African universities outside South Africa with high research productivity are the University of Ghana (in Ghana), Cairo University (in Egypt), and University of Gondar (in Ethiopia). Institutions outside Africa such as the London School of Hygiene and Tropical Medicine (in the United Kingdom (UK)), Oxford University (in the UK), and the Centers for Disease Control and Prevention (in the United States of America (USA)) are also major international contributors to vaccine research in Africa.

As shown in Figure 3, the most prolific authors of vaccine research publications from Africa are Madhi SA (South Africa), Wiysonge CS (South Africa), and Aaby P (Denmark and Guinea Bissau).

#### 3.3.2. The Most Prolific Institutions and Authors for Vaccine Trials in Africa

Figure 4 displays a bar chart ranking the top 20 institutions by the number of publications on vaccine trials from 2016 to 2024. The most productive African institutions for vaccine trials are the University of Witwatersrand, the University of Cape Town, and Setshaba Research Centre, in that order, all based in South Africa. The institutions located outside Africa that contribute the most to vaccine trial publications from Africa are the University of Oxford and the London School of Hygiene and Tropical Medicine (both in the UK) and the University of Maryland School of Medicine (in the USA).

The most prolific authors in vaccine trial publications from Africa are Madhi SA (South Africa), Aaby P (Denmark and Guinea Bissau), and Benn CS (Denmark and Guinea Bissau), as shown in Figure 5.

#### 3.3.3. The Most Prolific Institutions and Authors for Vaccine Reviews in Africa

The bar chart (Figure 6) ranks the top 20 institutions that have published reviews on vaccines from 2016 to 2024. The top African-based institutions in terms of publications of reviews on vaccines are the University of Witwatersrand (South Africa), the University of Cape Town (South Africa), and the University of Gondar (Ethiopia). The top non-African institutions that contributed the most to publication of reviews from Africa during the study period are the London School of Hygiene and Tropical Medicine, the University of Oxford, and the University of Edinburgh, all in the United Kingdom.

The most prolific authors in the publication of reviews on vaccines from Africa were Wiysonge CS (South Africa), Madhi SA (South Africa), and Le Doare K (UK and Uganda), as shown in Figure 7.

### 3.4. Distribution of Publications by Countries

#### 3.4.1. Distribution of Vaccine Research Publications by Corresponding Authors’ Countries

The top five African countries with the most publications on vaccine research were South Africa, Egypt Nigeria, Ethiopia, and Kenya (Figure 8). Figure 8 also shows both single-country publications and multiple-country publications, indicative of local and international collaborative research efforts. South Africa, which had the highest number of publications, had nearly equal numbers of multiple-country publications (MCPs) and single-country publications (SCPs). Egypt, Nigeria, and Ethiopia had slightly more SCPs than MCPs each, while Kenya had predominantly more MCPs than SCPs. Figure A1 presents a map showing the distribution of publications by country.

#### 3.4.2. Distribution of Vaccine Trials in Africa by Corresponding Authors’ Countries

All countries that conducted vaccine trials during the study period have more multiple-country publications than single-country publications, indicating strong international collaborations for clinical trials (Figure 9). South Africa, Egypt, Gambia, Kenya, and Guinea Bissau were the top countries publishing vaccine trials in Africa (Figure 9 and Figure A2).

#### 3.4.3. Distribution of Vaccine Reviews in Africa by Corresponding Authors’ Countries

This bar chart (Figure 10) shows the distribution of corresponding authors’ countries for vaccine reviews. Ethiopia leads with the highest number of publications, predominantly single-country publications. The next most prolific African countries are South Africa, Nigeria, Egypt, and Kenya, each with proportionally more multiple-country publications.

### 3.5. Trend Topics

#### 3.5.1. Trend Topics of Vaccine Research in Africa

The trend topics analysis of vaccine research from 2016 to 2024 (Figure A4) showed that before 2020, research concentrated on topics such as “vaccination”, “immunization schedule”, and “animals”, indicating a focus on general vaccine development, immunization programs, and preclinical studies. Methodological terms like “enzyme-linked immunospot assay”, “statistics and numerical data”, and “multivariate analysis” were also prominent, highlighting the use of immune assays and complex statistical methods to evaluate vaccine efficacy. In 2020, terms such as “COVID-19 vaccines”, “SARS-CoV-2”, “COVID-19 prevention and control”, and “COVID-19 epidemiology” became dominant. Additionally, population-specific terms such as “humans”, “child”, “female”, and “male” gained prominence. After 2020, COVID-19-related topics remained central, including “COVID-19 vaccines” and “SARS-CoV-2”, indicating ongoing efforts to refine and adapt vaccines for COVID-19. However, traditional vaccine research topics, such as “immunization schedule”, “vaccination”, and population-specific terms like “child” and “female” re-emerged, suggesting a return to broader vaccine research beyond COVID-19.

#### 3.5.2. Trend Topics of Vaccine Trials in Africa

The trend topics analysis of vaccine trials from 2016 to 2024 revealed distinct shifts in research focus. Before 2020, key topics included “Plasmodium falciparum”, “immunoglobulin G”, and “double-blind method”. Population-specific terms like “child”, “infant”, “female”, and “male” also featured prominently. In 2020, there was a marked shift towards terms such as “COVID-19 vaccines”, “SARS-CoV-2”, and “COVID-19 prevention and control”. Additionally, terms like “antibodies, viral” and “immunogenicity” showed increased frequency. After 2020, COVID-19 topics remained central, but research broadened to include continued attention to pediatric populations, as evidenced by the persistence of terms like “child” and “infant”. The emergence of “pregnancy” as a notable term point to growing interest in maternal vaccines.

#### 3.5.3. Trend Topics for Vaccine Reviews in Africa

The trend topics for vaccine reviews are somewhat like both the vaccine research and trials (Figure A6). Prior to 2020, research concentrated on public health and immunization topics such as “vaccination”, “statistics and numerical data”, and “infectious disease transmission, vertical/prevention and control”. Demographic terms like “humans”, “female”, “child”, and “infant” featured prominently, while regional terms like “Africa” and “global health”. Research on specific pathogens, including “*Streptococcus agalactiae*”, was also notable. In 2020, there was a clear shift toward COVID-19-related research, with terms such as “COVID-19 vaccines”, “SARS-CoV-2”, and “COVID-19 prevention and control” dominating. Public health behavior topics emerged, as seen in “health knowledge, attitudes, practice”, while demographic terms such as “pregnancy” and “child” remained significant. Post-2020, COVID-19 related reviews persisted, with terms like “COVID-19 vaccines/adverse effects” and “treatment outcome” being prominent.

## 4. Discussion

This bibliometric analysis of the landscape of vaccine and immunization research, trials and reviews in Africa provides an initial step toward understanding the evolution of research focus areas and the prioritization of vaccine-related studies across the continent.

Our analysis shows a strong annual growth rate of 55.39% in general vaccine research. However, vaccine trials showed a 6.08% decline during the same period. While the COVID-19 pandemic temporarily increased trial activity, this growth was short-lived, and by 2023, trial numbers had returned to pre-pandemic levels. This suggests that the pandemic’s influence on clinical trial output was transient, driven primarily by the global urgency around COVID-19 vaccines rather than by sustained improvements in Africa’s clinical trial infrastructure. Additionally, this current pace and scale are inadequate to meet the continent’s health needs [10,11]. The stagnation in trials highlights ongoing challenges in the vaccine research ecosystem, particularly in clinical trial capacity. Vaccine trials require more substantial infrastructure, funding, and regulatory frameworks than other types of research. Africa’s limited capacity to conduct large-scale, early-phase clinical trials is a key barrier that must be addressed. These challenges call for strategic investments in trial infrastructure, training of clinical research professionals, and regulatory harmonization across African countries.

The capacity for vaccine research in Africa has been a topic of concern and discussion for several years, requiring the need for sustained support and collaboration to strengthen it [10,12]. Our findings highlight the high degree of collaboration among researchers across the continent and internationally suggesting an area that can be tapped into for capacity building through strategic partnerships between more and less resourced countries and institutions. The African Vaccine Regulatory Forum (AVAREF) [13] positions itself in this regard, facilitating sustainable networks that could catalyze progress in vaccine research.

The higher number of vaccine research outputs in some countries compared to others maybe a reflection of broader economic and research capacity disparities across the continent. This disparity is attributed to a myriad of factors, including but not limited to insufficient research capacity, funding constraints, regulatory challenges, and lack of research infrastructure [11,14,15]. Additionally, studies have demonstrated that a country’s gross domestic product (GDP) is a significant predictor of its research output [14,15]. For example, in 2016, Nigeria, South Africa, and Egypt accounted for 65.7% of Africa’s overall expenditure on research and development [15]. In the same year, Egypt, Kenya, South Africa, Mali, Morocco, and Tunisia allocated 1% of their respective GDPs to research and development, whereas countries such as Lesotho invested less than 0.1% [15]. Despite this, there is still a notable lack of funding for research in Africa, which has been a continuous issue for a significant period [15]. This is an important aspect to consider when determining research priorities for the continent.

Addressing these disparities will require targeted investment in research infrastructure, funding, and capacity-building programs for countries with lower research outputs. Regional research networks can play a crucial role in sharing resources and expertise, enabling smaller countries to participate in research initiatives. Equitable funding mechanisms that prioritize under-resourced countries as well as different topics and types of research will be necessary to ensure that all African countries can contribute to and benefit from vaccine and immunization research. This then requires a better coordinated prioritization process at a regional level. However, the high levels of international co-authorship also highlight a potential overreliance on external funding and leadership, which may inadvertently limit local ownership, agenda-setting, and long-term sustainability of research efforts in Africa. In the context of shifting global priorities and increasing funding constraints, strengthening African-led research leadership will be essential, not only to ensure that research agendas reflect regional needs, but also to build resilient systems capable of driving innovation and sustaining progress in the face of external shocks. One promising step would be to establishment regionally coordinated expert consortia focused on vaccine and immunization research, which can serve as platforms for African-led priority setting, coordination, and oversight of collaborative studies.

The COVID-19 pandemic had a profound impact on vaccine research, leading to a surge in publications related to COVID-19 vaccines, epidemiology, and prevention. This shift highlights the adaptability of Africa’s research community in responding to urgent global health threats. However, the return to pre-pandemic levels of vaccine trial output in 2023 suggests that the momentum generated during the pandemic has not been sustained in broader vaccine research areas. While addressing COVID-19 was undoubtedly critical, the long-term research agenda must balance emerging threats with ongoing health challenges. Leveraging the infrastructure and collaborations built during the pandemic could help sustain research capacity for other vaccine-preventable diseases in Africa.

The growing focus on vaccine safety, particularly in relation to COVID-19 and malaria vaccines, represents a positive development in Africa’s vaccine research landscape. The pandemic has highlighted the importance of robust vaccine safety monitoring systems to ensure public confidence in new vaccines. As Africa rolls out new vaccines, such as those for malaria, it is essential to establish strong vaccine safety surveillance mechanisms across the continent. Building Africa-specific vaccine safety systems will be critical in maintaining public trust, especially in the face of vaccine hesitancy. These systems should be equipped to detect and respond to safety signals early, ensuring that new vaccines are both safe and effective for African populations. The COVID-19 experience offers valuable lessons that can be applied to strengthen vaccine safety monitoring for all vaccines in Africa.

## 5. Limitations

Our study’s reliance on PubMed as a single data source presents a limitation, as it may have missed some relevant publications indexed in other databases, including local studies that are not indexed in PubMed. Nevertheless, PubMed is renowned for its extensive collection of scientific publications in medical and biomedical sciences, which provided valuable and sufficient insights. In addition, the limit to 10,000 articles from PubMed for vaccine and immunization research may have led to some bias for this analysis, but we do not expect that this is substantial. Moreover, to avoid bias, these records were retrieved in chronological order and not based on specific filters or quality criteria.

Also, this bibliometric analysis could not delineate the specific type of research to enable us to differentiate whether they were biomedical, epidemiological, modeling, implementation or socio-behavioral in nature. As a result, certain stages of the research-to-policy pipeline may not be fully captured in our analysis, for example, the relative underrepresentation of implementation or social science research. More specific studies incorporating study type classification or classification potentially through machine learning approaches would provide a more nuanced understanding of research focus and potential gaps in the vaccine research landscape in the region. Study types such as systematic reviews and scoping reviews will be useful in filling this gap.

Finally, although we assessed international collaboration rates, our analysis did not differentiate author leadership roles (e.g., first or senior authorship), which could provide deeper insights into research ownership and capacity strengthening within Africa. Trials conducted in Africa but led exclusively by non-African institutions without any in-country affiliation may have been missed due to the structure of our search strategy, which focused on affiliations only.

## 6. Conclusions

In conclusion, while general vaccine and immunization research has grown significantly across the continent, the stagnation of clinical trials and disparities in research productivity among countries require urgent attention. Building on the momentum created during the COVID-19 pandemic, including the notable surge in evidence syntheses during that period, there is need for more attention on strengthening clinical trial infrastructure, fostering equitable collaborations, and developing research priorities to address the continent’s diverse health needs. Programmatic activities should focus on evidence-informed capacity building for researchers on the continent to conduct high-quality research and evidence syntheses that meet global standards, creating avenues for shared learning between countries, routine monitoring and tracking of progress made in the research landscape, and ensuring the link between conducting research and evidence synthesis is maintained, as this is key for ensuring research is efficiently translated into policy. The results of this study can guide the WHO AFRO, the African Union, and national governments in aligning funding and policy priorities with areas of demonstrated need, such as clinical trial capacity, under-researched disease areas, and country-level support mechanisms to advance a more equitable and sustainable vaccine research ecosystem across the continent.

## Figures and Tables

**Figure 1 vaccines-13-00509-f001:**
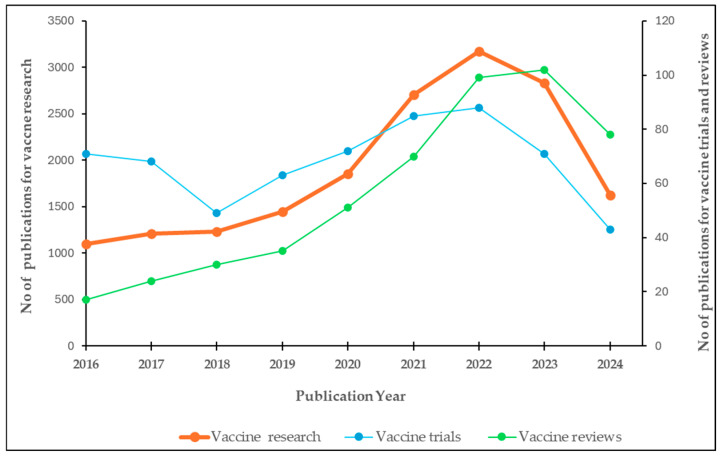
Annual publication trends in vaccine and immunization research, trials, and systematic reviews across Africa (2016–2024) *. The number of vaccine research articles (orange line) is plotted against the left Y-axis, while the number of trials (blue) and reviews (green) are plotted against the right Y-axis. * The number of articles in 2024 does not represent the total publications for the entire year, as the year was not yet over when the literature search was conducted.

**Figure 2 vaccines-13-00509-f002:**
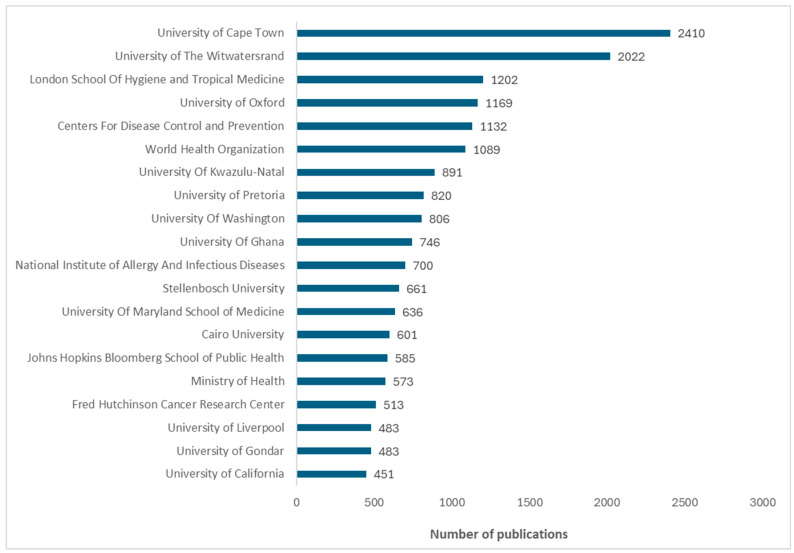
Top 20 institutions by number of publications on vaccine research from Africa (2016–2024).

**Figure 3 vaccines-13-00509-f003:**
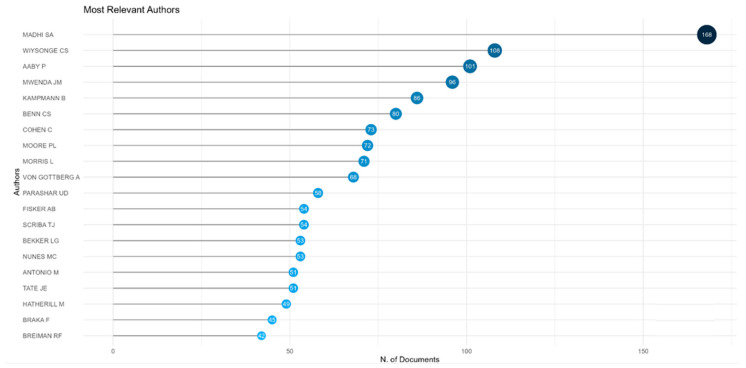
Top twenty prolific authors by number of publications on vaccine research from Africa 2016 to 2024.

**Figure 4 vaccines-13-00509-f004:**
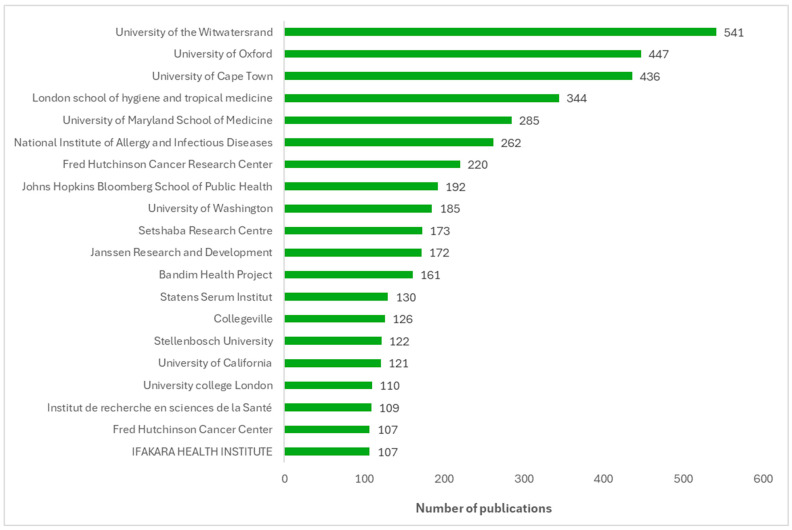
Top 20 institutions by number of publications on vaccine trials from Africa (2016–2024).

**Figure 5 vaccines-13-00509-f005:**
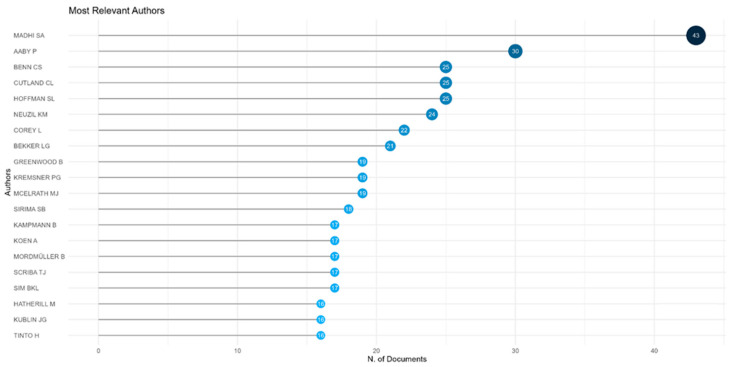
Top twenty prolific authors by number of publications on vaccine trials from Africa 2016 to 2024.

**Figure 6 vaccines-13-00509-f006:**
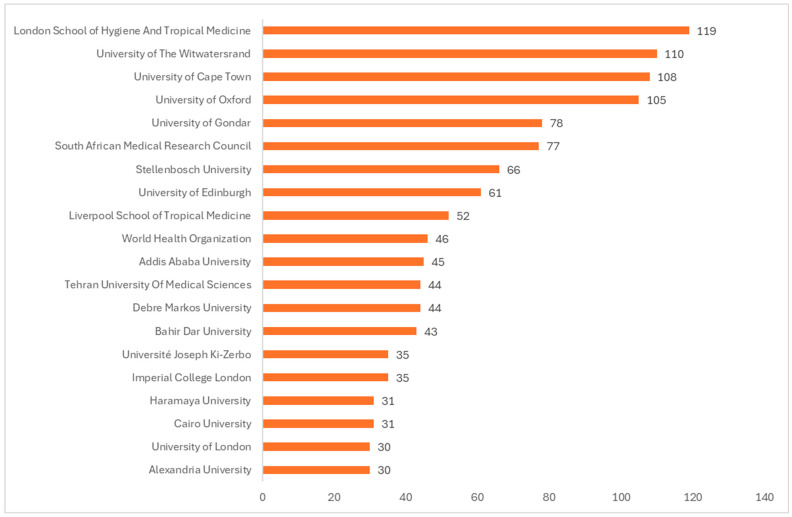
Top 20 institutions by number of publications on vaccine reviews from Africa (2016–2024).

**Figure 7 vaccines-13-00509-f007:**
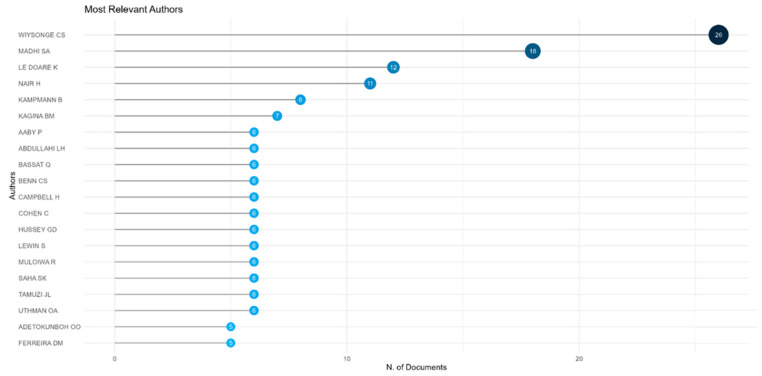
Top twenty prolific authors by number of publications on vaccine reviews from Africa from 2016 to 2024.

**Figure 8 vaccines-13-00509-f008:**
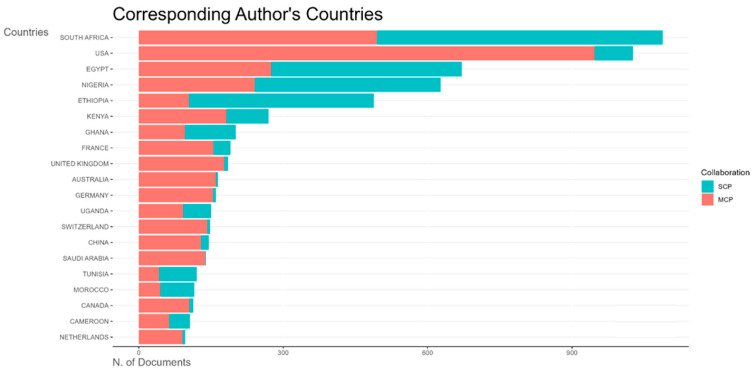
Top 20 most prolific countries by number of publications on vaccine research in Africa by collaboration type (SCP—single-country publication; MCP—multiple-country publication).

**Figure 9 vaccines-13-00509-f009:**
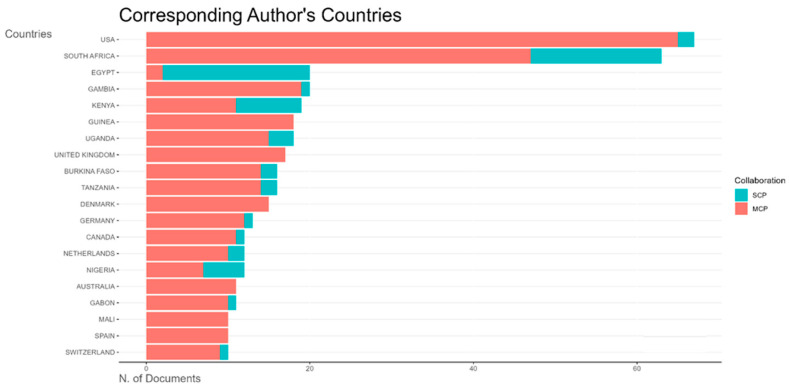
Top 20 most prolific countries by number of publications on vaccine trial publications in Africa by collaboration type (SCP—single-country publication; MCP—multiple-country publication).

**Figure 10 vaccines-13-00509-f010:**
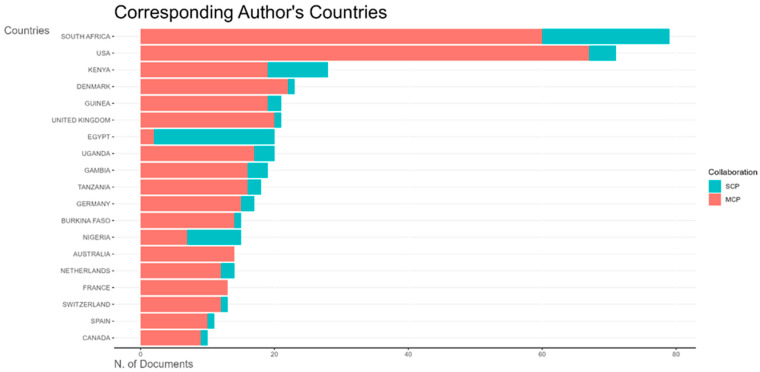
Top 20 most prolific countries by number of publications on vaccine reviews publications in Africa by collaboration type (SCP—single-country publication; MCP—multiple-country publication).

**Table 1 vaccines-13-00509-t001:** Summary of research publication metrics on vaccine research, trials, and reviews from Africa (2016–2024).

Description	Vaccine Research	Vaccine Trials	Vaccine Reviews
Number of articles	17,169	610	505
Sources (journals, books, etc.)	1565	149	209
Annual growth rate	55.39%	−6.08%	20.98%
Total number of authors	52,848	5866	3814
International co-authorship * (percentage)	65.83%	79.84%	68.6%

* Authors outside of Africa—International co-authorship is based on Bibliometrix’s default classification, defined as ≥1 author with a non-African institutional affiliation.

## Data Availability

Data generated for this review were all from publicly available publications.

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
