# Peer review of "Vaccine Research Trends in Africa from 2016 to Mid-2024: A Bibliometric Analysis"

_vaccines, 2025, doi:10.3390/vaccines13050509_

Round 1

Reviewer 1 Report

Comments and Suggestions for Authors

The topic of this paper is interesting. It is a bibliometric analysis of vaccine research in Africa that provides a valuable approach to assessing research trends, collaborations, and thematic focus areas. However, as the authors acknowledge, it has significant limitations. The main one is that the authors limit themselves to collecting information from PubMed without analyzing research quality criteria and without extracting information from each of the published papers.
Another limitation is that of the 17,169 papers found between 2016 and 2024, they only analyze the first 10,000. That is, 41.7% of the papers are excluded from the study. The question that arises is whether this significant percentage corresponds to the first years of searching in PubMed, so the bias is significant.
In this sense, conducting a systematic review or meta-analysis, including quality criteria for the papers analyzed, allows for more conclusive results. A more detailed analysis would allow for deeper conclusions to be drawn. There are some minor issues to keep in mind:
Line 114 and Table 1: The word "international" can be confusing. This is better, as the authors (authors outside of Africa) point out.
Figures 1A to 1C can be included in a single graph with different colors for each line.
The information included in sections 3.3.1, 3.3.2, and 3.3.3 can also be grouped under a single heading.

Author Response

RESPONSES TO REVIEWER 1

Comment 1: The topic of this paper is interesting. It is a bibliometric analysis of vaccine research in Africa that provides a valuable approach to assessing research trends, collaborations, and thematic focus areas. However, as the authors acknowledge, it has significant limitations. The main one is that the authors limit themselves to collecting information from PubMed without analyzing research quality criteria and without extracting information from each of the published papers.
Another limitation is that of the 17,169 papers found between 2016 and 2024, they only analyze the first 10,000. That is, 41.7% of the papers are excluded from the study. The question that arises is whether this significant percentage corresponds to the first years of searching in PubMed, so the bias is significant.
In this sense, conducting a systematic review or meta-analysis, including quality criteria for the papers analyzed, allows for more conclusive results. A more detailed analysis would allow for deeper conclusions to be drawn. There are some minor issues to keep in mind:
Line 114 and Table 1: The word "international" can be confusing. This is better, as the authors (authors outside of Africa) point out.
Figures 1A to 1C can be included in a single graph with different colors for each line.
The information included in sections 3.3.1, 3.3.2, and 3.3.3 can also be grouped under a single heading.

Response 1: We thank the reviewer for this insightful comment. We fully acknowledge the limitations of relying solely on PubMed for our analysis. However, PubMed remains one of the most comprehensive databases for peer-reviewed biomedical and public health literature as stated in our methods section (Page 4, lines 100-104). Regarding the limit to 10,000 records, we acknowledge this constraint imposed by the PubMed export functionality at the time of data retrieval (lines 399-403).

As suggested, we agree that further research such as systematic or scoping reviews would be valuable. We also stated this in the limitations section (lines 410-411).

Comment 2:

Line 114 and Table 1: The word "international" can be confusing. This is better, as the authors (authors outside of Africa) point out.

Response 2: Thank you. We added a footnote in Table 1, for more clarity (line 148)

Comment 3: Figures 1A to 1C can be included in a single graph with different colors for each line.
The information included in sections 3.3.1, 3.3.2, and 3.3.3 can also be grouped under a single heading.

Response 3: Thank you for this comment. Figures 1A-1C have now been grouped into one -Figure 1 (Line 153).

The information in  sections 3.3.1, 3.3.2, and 3.3.3 were actually grouped under the heading 3.3- Most prolific institutions and authors for vaccine research, trials and reviews (line 166-169)

Reviewer 2 Report

Comments and Suggestions for Authors

Peer Review of the Manuscript: “Vaccine research trends in Africa from 2016 to 2024: a bibliometric analysis”

Dear Authors,

I have carefully reviewed your manuscript, which investigates vaccine research trends in Africa over a nine-year period, using bibliometric techniques. Below are my observations and constructive suggestions for each section.

Title & Abstract

  • The title  reflects both the regional (Africa) and thematic (vaccine research) scope. However, it states “2016 to 2024,” yet you include data only up to August 2024. It might be worth clarifying in the title (e.g., “...to mid-2024”) or in the abstract that 2024 data are partial are there no studies from sept-dec 2024?.
  • Another potential improvement is to briefly mention that you used PubMed as the sole data source, as it has direct implications for the scope and representativeness of the results.
  • Consider adding one sentence summarizing the discrepancy between growth in general vaccine research versus clinical trials; that gap is a major study finding deserving more foregrounding in the abstract.

Introduction

  • The introduction compellingly highlights the importance of vaccine research and Africa’s disproportionate burden of vaccine-preventable diseases. Authors should clarify earlier why the time frame 2016–2024 was specifically chosen. You note a link to WHO AFRO’s transformation agenda (2015–2024), but a more explicit sentence about how that agenda guided your study period would be helpful.
  • You cite prior bibliometric analyses on African vaccine research and note that yours is broader in scope (encompassing general vaccine research, trials, and reviews).  You may wish to mention if there have been notable global bibliometric studies that include Africa, briefly contrasting them to your more Africa-focused approach.
  • The gap you identify e.g lack of up-to-date, continent-wide analyses on all vaccine types is logical. Emphasizing how your three-tiered analysis (general research, clinical trials, reviews) addresses that gap clarifies your contribution.

Materials and Methods

  • Relying solely on PubMed is understandable for medical literature, yet it inherently risks missing some regional or non-indexed publications. While you mention this limitation in your Discussion, consider adding a brief statement in the Methods explaining why PubMed was deemed sufficient (e.g., coverage, indexing standards).
    • You demonstrate how you built your search strings around African countries’ names and combined them with vaccine/immunisation terms. This is generally robust. However, you should clarify how you handled potential name variations or author affiliations not explicitly referencing an African country (e.g., “Harvard University, Senegal project”).
    • You note that for “vaccine research,” you only included the first 10,000 articles due to PubMed limit. It is important to specify how you decided on that subset (chronological order, relevance ranking, etc.) to mitigate concerns about selection bias.
  • The grouping into general vaccine research, vaccine trials, and vaccine reviews is helpful. One point that should be detailed is how you verified that each publication was indeed an African study or relevant to Africa (especially for multi-country collaborations).
  • The use of the Bibliometrix R package is well justified. Also, your mention of annual growth rates, top institutions, top authors, and collaboration networks demonstrates a comprehensive bibliometric approach.

Results

  • The breakdown of annual publications for each category (Figures 1A–1C) is clear, and you do a good job highlighting the COVID-19 “bump.” However, consider briefly noting whether the steep decline in 2023 (for trials, in particular) might be influenced by incomplete indexing of recent publications (i.e., there is often a lag in PubMed).
    • The bar charts are informative, but it would be good to clarify whether you combined or harmonized variations in institutional names (e.g., “Univ. of Cape Town” vs. “University of Cape Town”).
    • For top authors, you list affiliation in parenthesis. Ensure consistency and confirm that authors with multiple affiliations are accurately represented (e.g., “Denmark and Guinea Bissau”).
  • The single-country vs. multiple-country publication breakdown is quite useful. You’ve addressed the possibility of merging names (Congo vs. DRC, Guinea vs. Guinea-Bissau), but clarifying the approach in a single line in the text (not just in figure captions) would help avoid confusion.
  • These word clusters highlight historical shifts (pre- and post-COVID-19). The approach is valid. However, consider more explicit mention of how these trends were identified or operationalized (e.g., frequency thresholds).

Discussion

  • The authors provide a strong narrative around the mismatch between growing general vaccine research and stagnant clinical trials. This is a critical highlight. Authors should comment further on potential reasons for the decline in trials (regulatory or infrastructural barriers, reduced funding, etc.) even briefly since that is a central finding.
  • The surge in publications during COVID-19 is well noted. However, you may want to give a bit more nuance about whether the continent’s strong focus on urgent pandemic research overshadowed other vaccine trial efforts. This could strengthen your argument about needing more balanced, sustainable research investment.
  • You commendably link collaboration to capacity building. Since this is a major advantage for Africa (given the high international co-authorship rates), you might also underscore any potential pitfalls of overreliance on external funding or leadership in African studies and how that might affect local autonomy.
  • Limitations
    • You note the single-database limitation. Consider reinforcing it by mentioning that some local African journals may not be indexed in PubMed.
    • The 10,000-publication cap is described, but you might add a brief justification about how or why you anticipate minimal bias (e.g., the majority of high-impact, well-cited articles typically appear within the first portion, but this is not guaranteed).

Conclusion

  • You clearly call for strengthening clinical trial infrastructure, promoting equitable collaborations, and building capacity. Authors should possibly add one to two lines to emphasize how  how these findings can inform future policy or funding priorities at national/regional levels (e.g., the African Union, WHO AFRO).
  • Ensure that your conclusion addresses the three main streams of your study (general research, trials, and reviews). At times the conclusion focuses primarily on clinical trials, which is indeed the biggest gap, but re-mentioning key takeaways for reviews (e.g., the surge in COVID-19 review publications) would complete the final message.

Figures & Tables

  • Figure Quality & Labeling
    • Make sure all figures (especially Figures 2–10) have self-contained captions clarifying the population (e.g., “Top 20 institutions in Africa by number of publications on vaccine trials, 2016–2024”), including definitions of abbreviations (SCP, MCP).
  • Appendices: The maps and topic trend visuals are informative. Briefly verifying in the main text how readers should interpret “Appendix Figure 2a vs. 2b vs. 2c” would reduce confusion.

Comments on the Quality of English Language

The English could be improved to more clearly express the research.

Author Response

RESPONSES TO REVIEWER 2

Comment 1: Title & Abstract

  • The title  reflects both the regional (Africa) and thematic (vaccine research) scope. However, it states “2016 to 2024,” yet you include data only up to August 2024. It might be worth clarifying in the title (e.g., “...to mid-2024”) or in the abstract that 2024 data are partial are there no studies from sept-dec 2024?.
  • Another potential improvement is to briefly mention that you used PubMed as the sole data source, as it has direct implications for the scope and representativeness of the results.
  • Consider adding one sentence summarizing the discrepancy between growth in general vaccine research versus clinical trials; that gap is a major study finding deserving more foregrounding in the abstract.

Response 1: we thank you for these comments. We have updated the title and abstract accordingly, incorporating the use of PubMed as the sole data source in the abstract (line 20) and adding a sentence highlighting the discrepancy between vaccine research and clinical trials and its implications (lines 33-36).

Introduction

Comment2:

  • The introduction compellingly highlights the importance of vaccine research and Africa’s disproportionate burden of vaccine-preventable diseases. Authors should clarify earlier whythe time frame 2016–2024 was specifically chosen. You note a link to WHO AFRO’s transformation agenda (2015–2024), but a more explicit sentence about how that agenda guided your study period would be helpful.

  • You cite prior bibliometric analyses on African vaccine research and note that yours is broader in scope (encompassing general vaccine research, trials, and reviews).  You may wish to mention if there have been notable global bibliometric studies that include Africa, briefly contrasting them to your more Africa-focused approach.
  • The gap you identify e.g lack of up-to-date, continent-wide analyses on all vaccine types is logical. Emphasizing how your three-tiered analysis (general research, clinical trials, reviews) addresses that gap clarifies your contribution.

Response 2:

We thank you for these comments. We have revised accordingly:

  • On transformation agenda, please see-lines 93-98
  • Previous bibliometric analyses, please see lines 75-77

Materials and Methods

Comment 3: Relying solely on PubMed is understandable for medical literature, yet it inherently risks missing some regional or non-indexed publications. While you mention this limitation in your Discussion, consider adding a brief statement in the Methods explaining why PubMed was deemed sufficient (e.g., coverage, indexing standards).

Response 3: Thank you. We included this in our methods section, Lines 94-98

Comment 4: You demonstrate how you built your search strings around African countries’ names and combined them with vaccine/immunisation terms. This is generally robust. However, you should clarify how you handled potential name variations or author affiliations not explicitly referencing an African country (e.g., “Harvard University, Senegal project”).

Response 4: We appreciate the reviewer’s observation. Our search strategy was designed to capture studies that included vaccine or immunisation-related terms in combination with African country names. This approach also captured studies where African contexts were mentioned in the title, abstract, or keywords, even if the author affiliations did not explicitly reference an African institution. For instance, studies conducted in Africa but led by institutions such as “Harvard University” with no explicit African country in the affiliation were still likely to be included if the geographic context (e.g., Senegal) was mentioned elsewhere in the record. While we acknowledge that a few studies may have been missed due to indirect or absent geographic references, we believe such instances are minimal and unlikely to have significantly impacted the overall findings and trends reported in this analysis.

Comment 5: You note that for “vaccine research,” you only included the first 10,000 articles due to PubMed limit. It is important to specify how you decided on that subset (chronological order, relevance ranking, etc.) to mitigate concerns about selection bias.

Response 5: This was done in chronological order; we have included this in the limitations (417-418)

Comment 6: The grouping into general vaccine research, vaccine trials, and vaccine reviews is helpful. One point that should be detailed is how you verified that each publication was indeed an African study or relevant to Africa (especially for multi-country collaborations).

Response 6: Thank you for this important observation. We solely relied on our search strategy, which we believe to be robust. However, this presents a limitation as well, which has been acknowledged.

Comment 7: The use of the Bibliometrix R package is well justified. Also, your mention of annual growth rates, top institutions, top authors, and collaboration networks demonstrates a comprehensive bibliometric approach.

Response 7: Thank you very much for this comment.

Results

Comment 8: The breakdown of annual publications for each category (Figures 1A–1C) is clear, and you do a good job highlighting the COVID-19 “bump.” However, consider briefly noting whether the steep decline in 2023 (for trials, in particular) might be influenced by incomplete indexing of recent publications (i.e., there is often a lag in PubMed).

Response 8: This is well noted in the discussion: Lines 322-326

Comment 9: The bar charts are informative, but it would be good to clarify whether you combined or harmonized variations in institutional names (e.g., “Univ. of Cape Town” vs. “University of Cape Town”).

Response 9: We didn’t do this manually given the number of articles. We relied on the software to do this.

Comment 10: For top authors, you list affiliation in parenthesis. Ensure consistency and confirm that authors with multiple affiliations are accurately represented (e.g., “Denmark and Guinea Bissau”).

Response 10: Thank you for this important point. The software we used (Bibliometrix) counts author affiliations based on the frequency of occurrence within the metadata. In cases where authors have multiple affiliations (e.g., both Denmark and Guinea-Bissau), the software counts each country separately as they appear.

Comment 11: The single-country vs. multiple-country publication breakdown is quite useful. You’ve addressed the possibility of merging names (Congo vs. DRC, Guinea vs. Guinea-Bissau), but clarifying the approach in a single line in the text (not just in figure captions) would help avoid confusion.

Response 11: Thank you for this observation. Indeed, the search strategy adequately catered for this. We have included this statement in lines 106-109: “ To obtain country-specific publications, we listed all the countries in the African continent in our search strategy. This was helpful to ensure all countries were included and to precisely distinguish between countries with similar names (e.g., Congo versus Democratic Republic of Congo, Guinea versus Guinea-Bissau)” .

Comment 12: These word clusters highlight historical shifts (pre- and post-COVID-19). The approach is valid. However, consider more explicit mention of how these trends were identified or operationalized (e.g., frequency thresholds).

Response 12: These word clusters were automatically generated by the Bibliometric software.

Discussion

Comment 13: The authors provide a strong narrative around the mismatch between growing general vaccine research and stagnant clinical trials. This is a critical highlight. Authors should comment further on potential reasons for the decline in trials (regulatory or infrastructural barriers, reduced funding, etc.) even briefly since that is a central finding.

Response 13: Indeed, we agree, and we have highlighted these in the discussion-lines 322-326

Comment 14: The surge in publications during COVID-19 is well noted. However, you may want to give a bit more nuance about whether the continent’s strong focus on urgent pandemic research overshadowed other vaccine trial efforts. This could strengthen your argument about needing more balanced, sustainable research investment.

Response 14: Yes, we included this . Please see lines 322-341

Comment 15: You commendably link collaboration to capacity building. Since this is a major advantage for Africa (given the high international co-authorship rates), you might also underscore any potential pitfalls of overreliance on external funding or leadership in African studies and how that might affect local autonomy.

Response 15: Thank you for this comment. We have added some sentences based on your suggestion – line 371-377

Limitations

Comment 16: You note the single-database limitation. Consider reinforcing it by mentioning that some local African journals may not be indexed in PubMed.

Response 16: Thank you. We have included this. Please see lines 400-401

Comment 17: The 10,000-publication cap is described, but you might add a brief justification about how or why you anticipate minimal bias (e.g., the majority of high-impact, well-cited articles typically appear within the first portion, but this is not guaranteed).

Response 17: Thank you for this comment. The 10,000-publication cap was due to the export limitations of PubMed at the time of data retrieval. The records were retrieved in chronological order, covering the full study period from 2016 to 2024. While we acknowledge that this introduces a limitation, we anticipate minimal bias as the dataset still captures a representative spread of publication trends, authors, institutions, and themes over the entire time frame. Please see lines 410-411.

Conclusion

Comment 18: You clearly call for strengthening clinical trial infrastructure, promoting equitable collaborations, and building capacity. Authors should possibly add one to two lines to emphasize how these findings can inform future policy or funding priorities at national/regional levels (e.g., the African Union, WHO AFRO).

Response 18: A sentence has been added as suggested. Please see lines 434-438

Comment 19: Ensure that your conclusion addresses the three main streams of your study (general research, trials, and reviews). At times the conclusion focuses primarily on clinical trials, which is indeed the biggest gap, but re-mentioning key takeaways for reviews (e.g., the surge in COVID-19 review publications) would complete the final message.

Response 19: We thank you for this comment. We have revised accordingly to ensure these three components are adequately reflected.

Figures & Tables

Comment 20: Figure Quality & Labeling

Make sure all figures (especially Figures 2–10) have self-contained captions clarifying the population (e.g., “Top 20 institutions in Africa by number of publications on vaccine trials, 2016–2024”), including definitions of abbreviations (SCP, MCP).

Response 20: We have revised figure titles as suggested. The SCP and MCP abbreviations were defined in the text.

Appendices:

Comment 21: The maps and topic trend visuals are informative. Briefly verifying in the main text how readers should interpret “Appendix Figure 2a vs. 2b vs. 2c” would reduce confusion.

Response 21: Thank you for this comment. We have now added an explainer in the figure labels – Each bubble represents number of publications, with bubble size proportional to the number of associated publications”.

Reviewer 3 Report

Comments and Suggestions for Authors

1)The manuscript states that only the first 10,000 articles for vaccine research were analyzed due to PubMed's limit. It would strengthen the methodology to explain how this limit may have biased the data and whether stratified sampling or alternative databases (e.g., Scopus or Web of Science) were considered.

2)The analysis does not distinguish between biomedical, epidemiological, social science, or implementation research. This limitation is briefly mentioned, but further discussion of its implications on interpreting the findings would be beneficial.

3)While international collaborations are quantified, the roles of local versus international researchers (e.g., first or senior author status) are not explored. 

4)The manuscript rightly notes a publication surge during the pandemic, but it could benefit from exploring whether this translated into long-term capacity building or was a temporary redirection of global funding and attention.

Author Response

RESPONSES TO REVIEWER 3

Comment 1: The manuscript states that only the first 10,000 articles for vaccine research were analyzed due to PubMed's limit. It would strengthen the methodology to explain how this limit may have biased the data and whether stratified sampling or alternative databases (e.g., Scopus or Web of Science) were considered.

Response 1: Thank you for this comment. In our limitations we mentioned that we selected in chronological order  (please see lines 399-403)

Comment 2: The analysis does not distinguish between biomedical, epidemiological, social science, or implementation research. This limitation is briefly mentioned, but further discussion of its implications on interpreting the findings would be beneficial.

Response 2: Thank you for this comment. We have revised accordingly. Please see lines 403-410.

Comment 3: While international collaborations are quantified, the roles of local versus international researchers (e.g., first or senior author status) are not explored. 

Response 3: Thank you for this important observation. We have now noted this as a limitation in the discussion section. Please see lines 412-414

Comment 4: The manuscript rightly notes a publication surge during the pandemic, but it could benefit from exploring whether this translated into long-term capacity building or was a temporary redirection of global funding and attention.

Response 4: Indeed, we found that publications surged during the COVID-19 pandemic. We acknowledged this in our discussion . Pleases see lines 378-386;  

Round 2

Reviewer 1 Report

Comments and Suggestions for Authors

Suggestions have been included. The work may be published.

Author Response

Thank you very much for your comments. We appreciate

Reviewer 2 Report

Comments and Suggestions for Authors

Title & Abstract

  1. Hyphen error – “a bib-liometric analysis” still carries the split hyphen. Correct to “bibliometric”.
  2. Cut-off consistency – State the end-date once and identically in Title, Abstract and Methods (e.g., “1 January 2016 to 8 August 2024”).
  3. 10 000-record cap – Add a nine-word clause in the Abstract noting that growth-rate estimates use the first 10 000 PubMed records to pre-empt questions about selection bias.

Introduction

  • Line 72–76 claims no prior continent-wide work that includes trials and reviews. One recent global study (please cite, if appropriate) did include African trials, albeit not in detail. Amending that sentence will avoid over-statement.
  • Tighten the paragraph linking WHO-AFRO’s Transformation Agenda to the chosen study period: one well-placed sentence would suffice.

Materials & Methods

Issue

Required action

Affiliation-only filter

Acknowledge that trials conducted in Africa but led by non-African institutions with no in-country affiliation might be missed. One sentence in the Limitations is enough.

PubMed export ceiling

Note that the Entrez API can bypass the 10 000 limit; explain why the display-based export was retained (e.g., resource constraints, negligible difference in pilot tests).

Data cleaning

Specify the exact Bibliometrix routine (e.g., metaTagExtraction(db,"AU_UN")) used to harmonise institutional names and confirm that no manual standardisation was added.

Metric formulae

Provide the algebraic expression for Annual Growth Rate and clarify whether AGR for trials/reviews is based on their full counts or a truncated set for comparability.

International collaboration

State the rule (≥ 1 author with a non-African affiliation) and how dual affiliations were counted.

Results

  1. Figure 1 scaling – The dual-axis line compresses the trial/review trends. Either use separate panes or a log scale so smaller counts are visible.
  2. Bar charts – Add absolute numbers at the end of each bar (positions 18–20 differ by one paper).
  3. Country maps – Repeat, in the main text, that Congo/DRC and Guinea/Guinea-Bissau were merged so readers understand why those labels are absent.

Discussion

  • You now mention funding and regulatory hurdles for trials (lines 330-341) but cite only one generic review. Insert one Africa-specific example or reference (e.g., Alemayehu et al., 2018) to ground the statement.
  • The paragraph on over-reliance (lines 371-377) is excellent; consider proposing one concrete mitigation mechanism (e.g., mandatory local principal-investigator role in multicountry grants).

 Limitations

  • Add a brief note that using broad keyword/affiliation filters precludes finer study-type classification; future work could apply machine-learning tagging to distinguish implementation, social-science, modelling, etc.
  • Re-order the limitation sentences so the single-database issue appears first, followed immediately by the 10 000-record cap for coherence.

Conclusion

  • The new policy-orientation sentence (lines 434-438) is valuable. Echo, in one clause, the surge in COVID-19 evidence syntheses to ensure all three streams (research, trials, reviews) are reflected.

8 Figures & Tables

  • Define every abbreviation (AGR, SCP, MCP) within the figure captions themselves.
  • Confirm all images meet the journal’s minimum 300 dpi requirement; the embedded JPEGs appear at ~150 dpi.

References & Language

  • Correct “University of Edinburg” → “Edinburgh”.
  • Ensure consistent spelling (choose “immunization” or “immunisation” per journal style and apply throughout).
  • Scan once more for minor typos (e.g., double spaces line 305, “behaviour” vs “behavior”).
Comments on the Quality of English Language

 The English could be improved to more clearly express the research.

Author Response

Responses to reviewer’s comments

Title & Abstract

  1. Hyphen error – “a bib-liometric analysis” still carries the split hyphen. Correct to “bibliometric”.

Response- This has been corrected

  1. Cut-off consistency – State the end-date once and identically in Title, Abstract and Methods (e.g., “1 January 2016 to 8 August 2024”).

Response- This has been corrected

  1. 10 000-record cap – Add a nine-word clause in the Abstract noting that growth-rate estimates use the first 10 000 PubMed records to pre-empt questions about selection bias.

Response- This has been revised

Introduction

  • Line 72–76 claims no prior continent-wide work that includes trials and reviews. One recent global study (please cite, if appropriate) did include African trials, albeit not in detail. Amending that sentence will avoid over-statement.

Response: We have revised accordingly, citing two previously conducted studies. Please see lines 72-77

  • Tighten the paragraph linking WHO-AFRO’s Transformation Agenda to the chosen study period: one well-placed sentence would suffice.

Response: This has been revised . Please see lines 93-98

Materials & Methods

Issue

Required action

Affiliation-only filter

Acknowledge that trials conducted in Africa but led by non-African institutions with no in-country affiliation might be missed. One sentence in the Limitations is enough.

Response: This has been added to limitations-lines 431-434

PubMed export ceiling

Note that the Entrez API can bypass the 10 000 limit; explain why the display-based export was retained (e.g., resource constraints, negligible difference in pilot tests).

Response: Thank you for this insightful comment. We were not aware at the time of analysis that the Entrez API could be used to bypass the 10,000-record export limit on PubMed. We used the display-based export approach, which we considered appropriate based on the constraints of the web interface. Moving forward, we acknowledge the potential of using the Entrez API in future studies to enable retrieval of larger datasets and will consider this for more comprehensive analyses.

Data cleaning

Specify the exact Bibliometrix routine (e.g., metaTagExtraction(db,"AU_UN")) used to harmonise institutional names and confirm that no manual standardisation was added.

Response: Thank you for this observation. We analyzed institutional affiliation data based on the default outputs provided by Bibliometrix, without applying the metaTagExtraction(db, "AU_UN") function or any other affiliation-specific extraction routine. No manual standardization of institutional names was performed; the results reflect how affiliations appeared in the original PubMed metadata.

Metric formulae

Provide the algebraic expression for Annual Growth Rate and clarify whether AGR for trials/reviews is based on their full counts or a truncated set for comparability.

Response: Thank you for this comment. The Annual Growth Rate (AGR) was automatically computed by the Bibliometrix R package using its built-in growth function. The AGR for trials and reviews were based on full counts.we have updated the methods section- lines 127-131

International collaboration

State the rule (≥ 1 author with a non-African affiliation) and how dual affiliations were counted.

Response: International co-authorship was based on the default classification provided by Bibliometrix, which considers a publication international if at least one author has a non-African affiliation. Dual affiliations were not manually reviewed; the software includes any record listing a non-African institution under international collaboration.

We added a footnote in Table 1: International co-authorship is based on Bibliometrix's default classification, defined as ≥1 author with a non-African institutional affiliation. Lines 154-155

Results

  1. Figure 1 scaling – The dual-axis line compresses the trial/review trends. Either use separate panes or a log scale so smaller counts are visible.

Response: we used a foot note to describe the lines that belonged to either axis, which should be easy to interpret.

  1. Bar charts – Add absolute numbers at the end of each bar (positions 18–20 differ by one paper).

Response : We have added data labels to the bar charts

  1. Country maps – Repeat, in the main text, that Congo/DRC and Guinea/Guinea-Bissau were merged so readers understand why those labels are absent.

Response: We included this in methods: Due to software constraints in standardizing country names, certain countries with similar or overlapping spellings (e.g., Congo and Democratic Republic of Congo; Guinea and Guinea-Bissau) were merged during analysis, resulting in combined article counts. Lines 135-138

Discussion

  • You now mention funding and regulatory hurdles for trials (lines 330-341) but cite only one generic review. Insert one Africa-specific example or reference (e.g., Alemayehu et al., 2018) to ground the statement.

Response- We have added the reference (line 345)

  • The paragraph on over-reliance (lines 371-377) is excellent; consider proposing one concrete mitigation mechanism (e.g., mandatory local principal-investigator role in multicountry grants.

Response: We have done so. Lines 387-390

 Limitations

  • Add a brief note that using broad keyword/affiliation filters precludes finer study-type classification; future work could apply machine-learning tagging to distinguish implementation, social-science, modelling, etc.

Response: We have included this in the sentence. Please see lines 425-427

  • Re-order the limitation sentences so the single-database issue appears first, followed immediately by the 10 000-record cap for coherence.

Response: This was done as suggested

Conclusion

  • The new policy-orientation sentence (lines 434-438) is valuable. Echo, in one clause, the surge in COVID-19 evidence syntheses to ensure all three streams (research, trials, reviews) are reflected.

Response: Thus was added. See lines 441-442

8 Figures & Tables

  • Define every abbreviation (AGR, SCP, MCP) within the figure captions themselves.

Response: Abbreviations for SCP and MCP were added to the figures. Please see lines 268,279, and 289. We did not use the abbreviation AGR in Table 1

  • Confirm all images meet the journal’s minimum 300 dpi requirement; the embedded JPEGs appear at ~150 dpi.

Response: We confirm that.

References & Language

  • Correct “University of Edinburg” → “Edinburgh”.

Response: Done

  • Ensure consistent spelling (choose “immunization” or “immunisation” per journal style and apply throughout).

Response: Done

  • Scan once more for minor typos (e.g., double spaces line 305, “behaviour” vs “behavior”).

Response: Done